# Recent Advances in Pathology of Intrahepatic Cholangiocarcinoma

**DOI:** 10.3390/cancers16081537

**Published:** 2024-04-17

**Authors:** Joon Hyuk Choi, Swan N. Thung

**Affiliations:** 1Department of Pathology, Yeungnam University College of Medicine, Daegu 42415, Republic of Korea; 2Department of Pathology, Molecular and Cell-Based Medicine, Icahn School of Medicine at Mount Sinai, 1468 Madison Avenue, New York, NY 10029, USA; swan.thung@mountsinai.org

**Keywords:** intrahepatic cholangiocarcinoma, pathology, diagnosis, molecular genetics, classification

## Abstract

**Simple Summary:**

Intrahepatic cholangiocarcinoma (ICCA) is the second most common type of primary liver malignancy after hepatocellular carcinoma (HCC). ICCA is characterized by molecular heterogeneity and a diverse histopathological spectrum. Generally, the prognosis for patients diagnosed with ICCA is poor. Recent advances have improved our understanding of the molecular genetics and histological subtypes of ICCA. An accurate diagnosis of ICCA is important for patient management and prognosis. This review aims to provides an updated overview of the pathology of ICCA, with a particular focus on its molecular genetics, histological subtypes, and the diagnostic approaches necessary to distinguish it from other diseases.

**Abstract:**

Intrahepatic cholangiocarcinoma (ICCA) is a malignant epithelial neoplasm characterized by biliary differentiation within the liver. ICCA is molecularly heterogeneous and exhibits a broad spectrum of histopathological features. It is a highly aggressive carcinoma with high mortality and poor survival rates. ICCAs are classified into two main subtypes: the small-duct type and large-duct types. These two tumor types have different cell origins and clinicopathological features. ICCAs are characterized by numerous molecular alterations, including mutations in *KRAS*, *TP53*, *IDH1/2*, *ARID1A*, *BAP1*, *BRAF*, *SAMD4*, and *EGFR*, and *FGFR2* fusion. Two main molecular subtypes—inflammation and proliferation—have been proposed. Recent advances in high-throughput assays using next-generation sequencing have improved our understanding of ICCA pathogenesis and molecular genetics. The diagnosis of ICCA poses a significant challenge for pathologists because of its varied morphologies and phenotypes. Accurate diagnosis of ICCA is essential for effective patient management and prognostic determination. This article provides an updated overview of ICCA pathology, focusing particularly on molecular features, histological subtypes, and diagnostic approaches.

## 1. Introduction

Cholangiocarcinoma (CCA) is an aggressive malignant neoplasm with biliary differentiation that arises along the biliary tree [1]. Based on their anatomical location, CCAs are divided into three categories: intrahepatic, perihilar, and distal CCA. Intrahepatic CCA (ICCA) is the second most prevalent type of primary liver malignancy (after hepatocellular carcinoma (HCC)), accounting for approximately 10–15% of all cases [2]. The incidence of ICCA has increased in several geographical locations [3,4]. In the United States, the incidence of ICCA has increased from 0.92 cases per 100,000 person-years between 1995 and 2004 to 1.09 cases per 100,000 person-years between 2005 and 2014 [5].

ICCA represents a heterogeneous group of tumors that are diverse in terms of clinical presentation, pathological features, and molecular characteristics. This diversity is closely related to the origin of the cells, pathogenesis, the presence of underlying liver diseases, and molecular alterations [6]. ICCAs exhibit aggressive behavior, with high mortality and poor survival rates [7,8]. Table 1 summarizes the evolution of the WHO classification of ICCA.

According to the World Health Organization (WHO) classification of digestive system tumors (5th edition), ICCAs have two main subtypes: small-duct and large-duct types [1]. Small-duct-type ICCA occurs in the peripheral parts of the liver, and is also called the peripheral type. The prevalence of this subtype is regionally dependent and accounts for approximately 40–90% of ICCAs. Large-duct-type ICCA arises in the larger intrahepatic bile ducts near the hepatic hilum, close to the right and left hepatic ducts, and is also called the hilar type and perihilar type. These two types differ in their etiologies, clinical behavior, and pathological features and have different genetic alterations [9,10]. In the future, it will be necessary to incorporate these findings into clinical research and study processes.

Recently, molecular genetics approaches and omics-based integrative studies have helped us better understand the biology, classification, prognosis, and treatment of ICCAs [11,12,13,14,15]. Precise pathological classification and understanding of the molecular pathogenesis of ICCA are essential for efficient patient management and prognostication. However, due to its histology and molecular genetics variability, ICCA is still difficult to diagnose and treat. This review provides an update on ICCA pathology, mainly focusing on changes in its molecular genetics, various histological subtypes, and diagnostic approaches, which distinguish it from other conditions.

## 2. Risk Factors

The etiologies of most CCAs remain unclear. Nonetheless, several risk factors for CCA exist, and their prevalence has significant variability across different regions [1]. Risk factors for ICCA depend on the tumor location. Small-duct ICCA has the same risk factors as HCC, which include chronic viral hepatitis (hepatitis B and hepatitis C), cirrhosis (regardless of the cause), alcoholic liver disease, hemochromatosis, metabolic syndrome, diabetes mellitus, obesity, and non-alcoholic fatty liver disease (NAFLD)/non-alcoholic steatohepatitis (NASH). The risk factors for large-duct ICCA are similar to those for extrahepatic and perihilar CCA, which include liver fluke infections (*Clonorchis sinensis* and *Opisthorchis viverrini*), primary sclerosing cholangitis (PSC), and hepatolithiasis [16,17]. Other risk factors include malformations of the biliary tract (such as Caroli disease, congenital hepatic fibrosis, bile duct cyst, and choledochal cyst) [18,19] and chemical and occupational exposures (such as Thorotrast, asbestosis, smoking, 1,2-dichloropropane, and dichloromethane) [20]. However, the exact mechanisms underlying ICCA tumorigenesis are yet to be clarified.

## 3. Cells of Origin

The two main histological subtypes of ICCAs, the small- and large-duct types, are thought to arise from different cell types [21]. Small-duct ICCA exhibits characteristics similar to those of mucin-negative cuboidal cholangiocytes, including the presence of hepatic progenitor cells (HPCs). HPCs can differentiate into both hepatocytes and cholangiocytes and are found in the Canals of Hering [22]. Therefore, HPCs have been proposed as the origin of small-duct ICCA [23,24]. However, small-duct ICCAs can also originate from transformed and transdifferentiated HPCs and mature hepatocytes [25,26]. No specific precursor lesion has yet been identified in small-duct ICCA; however, there have been rare cases of biliary adenofibroma [27,28].

Large-duct ICCA has strong similarities to perihilar CCA in terms of immunohistochemistry, gene expression profiles, growth patterns, and the presence of preneoplastic lesions in cholangiocytes and peribiliary glands (PBGs). Biliary tree stem/progenitor cells located in the PBGs have been suggested as candidate cells for the origin of large-duct ICCA [29]. Large-duct ICCA can develop from three different types of premalignant intraductal lesions: biliary intraepithelial neoplasia (BilIN), intraductal papillary neoplasm of the bile ducts (IPNB), and intraductal tubulopapillary neoplasm of the bile ducts (ITPN) [30,31].

## 4. Clinical Features

ICCA typically occurs in elderly patients. The mean age at the time of ICCA diagnosis is more than 50 years, with the highest number of cases appearing in patients in their 50s and 70s. Men are slightly more likely to be affected by ICCA than women [1]. The clinical features of ICCA vary depending on the anatomical location, growth pattern, and tumor stage. Small-duct ICCAs often do not cause any symptoms until they reach a relatively large size. General malaise, nausea, abdominal pain in the right upper quadrant, night sweats, and weight loss are the common symptoms. Large-duct ICCAs that cause obstruction of the central bile duct may manifest symptoms such as jaundice or cholangitis.

Altered liver function test results—including elevated total and direct bilirubin, alkaline phosphatase, and gamma-glutamyl transferase—are often observed at the time of ICCA diagnosis but are not specific [11]. Serum CA19-9 levels are typically elevated but have limited diagnostic utility because of their low sensitivity and specificity [10]. Recently, it was reported that patients with ICCA had higher serum levels of doublecortin-like kinase 1 (DCLK1) than those with HCC, and that DCLK1 was undetectable in healthy individuals [32]. Therefore, DCLK1 could serve as a serum biomarker for the early diagnosis of ICCA.

## 5. Radiological Features

Imaging methods—including ultrasonography, computed tomography (CT), and magnetic resonance imaging (MRI)—are essential tools for managing ICCA, as they are useful in diagnosis, the determination of tumor stage, monitoring progress, and evaluating the effectiveness of treatment [33]. The diagnostic accuracy of these imaging techniques is influenced by the anatomical location and growth patterns of the ICCA [34]. CT is regarded as the standard imaging technique for the preoperative assessment of ICCA, as it can thoroughly assess the primary tumor, its relationship with adjacent structures, and its extension into the thoracic and abdominal regions [34]. The accuracy of MRI is comparable to that of CT in terms of diagnosing and staging ICCA.

On CT and MRI images, ICCA typically exhibits peripheral rim enhancement during the arterial phase. Subsequently, progressive and uniform accumulation of the contrast agent occurs in the late phase, leading to homogeneous enhancement [35,36]. In ICCA, a targetoid pattern featuring arterial rim enhancement, peripheral washout, or delayed central enhancement may also be detected [37]. ^18^F-fluorodeoxyglucose positron emission tomography (^18^FDG-PET) may be particularly valuable for evaluating the presence of lymph nodes or distant metastases [38].

## 6. Molecular Features

ICCAs exhibit notable genetic diversity associated with different causative factors. The frequency of reported genetic mutations in ICCA varies and is influenced by factors such as ethnic backgrounds, diagnostic methods, various risk factors, and the anatomical location of the tumor in the biliary system [39]. Recent studies have identified many molecular alterations in ICCAs. These include mutations in *KRAS*, *TP53*, *IDH1/2*, *ARID1A*, *BAP1*, *BRAF*, and *EGFR* [1]. In addition, mutations in genes related to oncogenic signaling pathways, such as *PIK3CA* and *MET*, and various gene fusions, particularly those involving *FGFR2*, have been observed in ICCAs [40,41,42].

Mutations in *IDH1/2* and *BRAF* and *FGFR2* fusions are observed exclusively in small-duct ICCA [43,44]. Conversely, mutations in *KRAS, SMAD4,* and *TP53* and the amplification of *MDM2* are more frequently observed in large-duct ICCA [44,45,46,47,48]. Genetic and epigenetic alterations associated with inflammation have been described in large-duct ICCA. For instance, COX-2 expression is highly elevated in ICCAs associated with primary sclerosing cholangitis (PSC) [49]. *KRAS* mutations have been suggested to be an early event in cholangiocarcinogenesis associated with PSC and hepatolithiasis [47,50]. In patients with ICCA, seropositivity for hepatitis B surface antigen (HBsAg) has been linked to *TP53* mutations [51]. Approximately 9% of liver fluke-associated CCAs exhibit activation mutations in *GNAS* [52]. Mismatch repair (MMR) deficiency is observed in approximately 6% of ICCA cases [53]. Patients with MMR deficiency often exhibit a solid, mucinous, or signet ring cell appearance [54]. Whole-genome expression and mutation analyses have classified ICCA into two main molecular subtypes: the proliferation subclass and inflammation subclass [26,42,55,56,57,58,59,60,61].

## 7. Immunohistochemical Features

The classification of small and large-duct ICCAs is based on morphological features, and immunohistochemistry (IHC) may not be necessary for the diagnosis of ICCA. However, in cases with poorly differentiated or hybrid morphologies, classification based solely on morphology is challenging. IHC is useful for determining the lineage of tumor cell differentiation. Small- and large-duct ICCAs exhibit similarities in many immunohistochemical markers, such as cytokeratin (CK)7, CK19, and MUC1. However, only a few markers are specific to both types [1]. C-reactive protein (CRP) is a marker with high sensitivity and specificity for detecting small-duct ICCA, showing positivity in 95% of small-duct ICCA cases. In contrast, it is positive in only 5% of large-duct ICCA cases [62,63]. N-cadherin is another marker with high sensitivity and moderate specificity. It is positive in 87% of small-duct ICCA cases, but positivity is only shown in 16% of large-duct ICCA cases [62]. Tubulin beta-III (TUBB3) and CD56 (NCAM) can also be positive for small-duct ICCA, but their sensitivity and specificity are not as high as those of CRP and N-cadherin [64].

S100 calcium-binding protein P (S100P) is a useful biomarker for the identification of large-duct ICCA. Various antibody clones specific for S100P are commercially available, and their positivity rates may differ among them. In a prior study that evaluated three distinct clones, a monoclonal antibody (clone 16/S100P) demonstrated the most reliable performance, being positive in 95% of large-duct ICCA cases compared with 29% of small-duct ICCA cases [62]. Large-duct ICCA commonly exhibits a loss of SMAD4 expression, whereas a loss of BAP1 expression is predominantly found in small-duct ICCA [65]. Albumin mRNA in situ hybridization (ISH) is frequently positive in ICCA [66]. Albumin was expressed more frequently in small-duct ICCA (71%) than it was in large-duct ICCA (18%) [67].

## 8. Pathological Features of Conventional Intrahepatic Cholangiocarcinoma

ICCAs are grossly classified into four types: the mass forming (MF), periductal infiltrating (PI), intraductal growth (IG), and mixed types. Small-duct ICCA generally shows the MF type, whereas large-duct ICCA usually shows the PI and PI + MF type. The MF and PI patterns show a similar incidence among ICCAs [9,43]. People with mixed MF and PI types often have a worse prognosis than those with other types of ICCA [68]. ICCAs are predominantly adenocarcinomas. In most cases, they have a ductal or tubular pattern with lumens of varying sizes. There is a variable and often abundant fibrous stroma [43,69]. The tumor cells are generally small or medium in size, exhibiting a cuboidal or columnar shape, and they may display pleomorphism. Most tumor cells have a pale, slightly eosinophilic, clear, or vacuolated cytoplasm. The nuclei are typically smaller, and the nucleoli are usually less conspicuous than those in HCC.

### 8.1. Small-Duct Intrahepatic Cholangiocarcinoma

Small-duct ICCAs appear mainly as whitish or gray, firm, nodular masses in the liver parenchyma (an MF pattern) (Figure 1a). The tumor is well demarcated but lacks a surrounding fibrous capsule. The tumor size varies considerably among cases, ranging from microscopic lesions to large bulky masses over 10 cm. They are usually solid; however, cystic degeneration of variable sizes may occur [70]. Satellite nodules are also common.

Histologically, small-duct ICCA shows tubular formations with distinct lumens consisting of small- to medium-sized, cuboidal to low columnar tumor cells with scant cytoplasm (Figure 1b). Microtubular, solid, micropapillary, anastomosing trabecular, cord-like growth patterns, or spindle cell nests with slit-like lumens may be variably present [9,43,69,71]. Mucin secretion is usually absent or scanty. An abundant fibrotic stroma is observed. Small-duct ICCA shows replacement growth of tumor cells in the hepatic lobules or regenerative nodules. In advanced cases of small-duct ICCA, solid growth may be observed at the tumor periphery, with the central areas exhibiting extensive sclerosis and hypovascularity.

### 8.2. Large-Duct Intrahepatic Cholangiocarcinoma

Large-duct ICCAs appear as periductal nodular and sclerosing tumors extending along the bile duct wall (PI pattern) (Figure 2a). The tumor causes thickening, stricture, or obliteration of the affected large bile ducts. If the tumor invades the adjacent liver parenchyma, it may present as a nodular mass. The tumor may exhibit extensive lateral spread to the perihilar or distal bile ducts. Mixed periductal-infiltrating and mass-forming patterns can be observed. In advanced stages, ICCAs comprise nodules of different sizes that often merge.

Histologically, large-duct ICCA resembles perihilar and extrahepatic CCA [43,72]. It is an invasive tubular adenocarcinoma consisting of intermediate to large glands characterized by tall columnar cells (Figure 2b). The tumor shows a desmoplastic reaction that invades the portal connective tissue, adjacent bile ducts, and hepatic parenchyma. Mucus secretion is common. Large bile ducts from which it originates often exhibit sclerosis or obliteration due to tumor tissue [9,69,72]. Compared with small-duct ICCAs, large-duct ICCAs are characterized by the formation of irregular and angulated glands that infiltrate a desmoplastic stroma and exhibit abundant cytoplasmic and intraluminal mucin production. In poorly differentiated large-duct ICCAs, scattered nests of pleomorphic cancer cells are present. Lymphovascular and perineural invasion and lymph node metastases are common. In large-duct ICCAs, premalignant lesions such as BilIN and IPNB are often observed in the adjacent ducts [1]. Large-duct ICCA may display a morphology resembling that of the small-duct type at the interface between the cancer and liver tissue at the invasive front [43,45]. Similarly, small-duct ICCA may have small foci of a morphology resembling that of the large-duct type. Table 2 shows a comparison of the small-duct type and large-duct types of conventional ICCA.

### 8.3. Histological Grading

No definitive criteria for the histological grading of ICCAs have been established for ICCAs. In the 2019 WHO classification of liver tumors, ICCAs are graded as well-, moderately, or poorly differentiated adenocarcinoma based on their morphology, using a three-tiered system [1]. The College of American Pathologists (CAP) has proposed the following quantitative histological grading system based on the extent of gland formation within the tumor: grade X, the grade cannot be assessed; grade 1, well differentiated (more than 95% of the tumor composed of glands); grade 2, moderately differentiated (50–95% of the tumor composed of glands); and grade 3, poorly differentiated (less than 49% of the tumor composed of glands) [73]. The histologic grading depends on the morphology and extent of gland formation. Generally, grading a neoplasm requires morphologic variation within a given tumor. For example, because signet ring cell carcinoma has little variation, there is no practical way to grade this type of carcinoma. Rare ICCA subtypes with no gland formation include lymphoepithelioma-like carcinoma (LELCCA), sarcomatous ICCA, and acinar cell carcinoma, whereas ICCA subtypes with glandular formation include adenosquamous carcinoma. The undifferentiated category is rarely used and is designated for tumors lacking any obvious glandular, squamous, or neuroendocrine differentiation based on their morphology and immunohistochemistry.

### 8.4. Premalignant Lesions of Intrahepatic Cholangiocarcinoma

BilIN, IPNB, and ITPN are precursors of large-duct ICCA, whereas the precursors of small-duct ICCA are unknown. BilIN is a microscopic, non-invasive, flat, or micropapillary lesion that is confined to the bile ducts [30,31,45,69,74]. BilIN is characterized by a dysplastic epithelium with multilayered nuclei. Based on the extent of cellular and nuclear atypia, BilINs are divided into low-grade and high-grade. IPNB is a grossly visible premalignant neoplasm with intraductal papillary or villous growth of the biliary-type epithelium [75]. IPNBs are categorized into types 1 and 2, according to their similarity to their counterparts found in the pancreas. Type 1 IPNBs histologically resemble the intraductal papillary mucinous neoplasms of the pancreas, whereas type 2 IPNBs differ to varying extents. Type 1 IPNBs are more common in the intrahepatic bile ducts, whereas type 2 IPNBs are more common in the extrahepatic bile ducts. Invasive carcinomas occur more frequently in type 2 IPNBs than in type 1 IPNBs [76,77]. ITPN is characterized as a preinvasive, mass-forming intraductal neoplasm of the intra- or extrahepatic bile ducts, consisting predominantly of nonmucinous tubular structures with and without sheet-like growth (≥70% of the neoplasm) and with no or only minimal papillary growth [78,79]. The majority of ITPNs are found to be associated with invasive carcinoma at the time of diagnosis.

Premalignant lesions of unconventional ICCAs are not known. Unconventional ICCAs usually develop on the background of a nonbiliary chronic liver disease and cirrhosis. The malignant transformation in bile duct adenoma is considered to be extremely low [80]. The presence of a high frequency of *BRAF* V600E mutations in bile duct adenomas suggests that they are true neoplasms and that they may be important precursors for the subset of ICCA that harbor *BRAF* mutations [81].

## 9. Rare Subtypes of Intrahepatic Cholangiocarcinoma

Although most intrahepatic biliary malignancies are adenocarcinomas characterized by a conventional ductal or tubular morphology, the 2019 WHO classification has recognized various histological subtypes of ICCA [1]. ICCA subtypes are rare, each accounting for less than 5% of all ICCAs [70]. The classification of subtypes is primarily based on distinctive histological features, and it is still unclear whether or not these subtypes exhibit unique molecular characteristics [70]. Some subtypes are specific to either the small- or large-duct type. In particular, cholangiolocarcinoma and ICCA with a ductal plate malformation pattern are categorized as subtypes of small-duct ICCA. These subtypes can develop independently or in association with conventional ICCA. Other rare subtypes are not yet completely separated from the small- and large-duct ICCA types. It is essential to identify the histological subtypes because they have a better or worse prognosis than that of conventional CCA.

### 9.1. Cholangiolocarcinoma

Cholangiolocarcinoma (also called cholangiolocellular carcinoma) is a subtype of ICCA characterized by a ductular configuration (>80% of the tumor) [1]. The cholangiolocellular pattern was first described in 1959 as a tumor that could be deceptively bland, mimicking bile ductular proliferation [82]. This tumor was categorized according to the 2010 WHO classification as a combined HCC-CCA with stem cell features, and as the cholangiollocellular type [83]. However, based on morphological, immunohistochemical, and molecular similarities, this tumor has been categorized as a subtype of small-duct ICCA according to the 2019 WHO classification [1]. At least 50% of patients have a history of chronic liver diseases, such as viral hepatitis and hemochromatosis [25]. Cholangiolocarcinoma shares clinical and imaging features with both HCC and ICCA [70]. The 5-year survival rate is 75%, which is significantly better than that of patients with conventional ICCA [84].

Histologically, the tumor cells are smaller cuboidal cells with round to oval nuclei with fine chromatin and scant cytoplasm (Figure 3a,b). The tumor cells show minimal to mild atypia. The tumor is characterized by anastomosing cords and glands of tumor cells with low-grade nuclei and an “antler horn–like” branching pattern, mimicking a ductular reaction. An anastomosing pattern refers to cancer cells forming a network-like structure that branches and reconnects similarly to cholangiolar structures. This growth pattern is commonly present in the ICCA subtype, particularly in cholangiolocarcinoma. Hyalinized fibrotic stroma is present. Mucin production is usually absent. Necrosis is uncommon, even in cases with large tumors. Immunohistochemically, the tumor cells are positive for CD56 (NCAM) and show luminal expression of epithelial membrane antigen (EMA). CD56 (NCAM) is known to be positive in bile ductules but negative in bile ducts in the non-neoplastic liver and is useful for determining the cholangiolocellular phenotype. However, it is not always expressed in cholangiolocarcinoma (positive in 70–80%) and can be positive in conventional CCA (10–20%) [70].

Differential diagnosis of cholangiolocarcinoma includes the ductular reaction. The ductular reaction can mimic cholangiolocarcinoma [85]. Marked background inflammation, lobular architecture, minimal nuclear atypia, and lack of lymphovascular or perineural invasion favor the ductular reaction [86]. The presence of cytological and architectural atypia, an infiltrative growth pattern, a high Ki-67 index, diffuse strong p53 staining, BAP1 loss, and genomic alterations, such as *IDH1/IDH2* mutation and *FGFR2* fusion, favors ICCA [9].

### 9.2. ICCA with Ductal Plate Malformation

ICCA with ductal plate malformation is a rare subtype of ICCA that shows a ductal plate malformation pattern in more than 50% of the tumor area [1,69]. The microscopic features are reminiscent of ductal plate malformation in congenital hepatic fibrosis, Caroli disease, and polycystic liver [87]. According to a 10-case series, 60% of patients had a history of chronic liver disease. Tumors are usually less than 5 cm in size. *FGFR2* and *PTPRT* are the most frequently mutated genes in this subtype [88]. This subtype is frequently intermixed with HCC or conventional CCA components [70]. The prognosis of ICCA with a DPM pattern has not yet been studied.

Histologically, the tumor shows a vague multinodular architecture with intervening fibrous stroma [70]. More than half of the tumor area shows a ductal plate malformation pattern (Figure 4a,b). The tumor cells appear benign-looking, small, oval, and non-pleomorphic. The tumor cells are arranged in irregularly dilated ducts with distinctive projections and bridges. The portal tracts and central veins are regularly distributed throughout the tumor, suggesting a replacing growth pattern. Bile plugs are frequently present in dilated neoplastic 1umens [1]. Mitotic figures are rare. The MIB-1 index is usually less than 5%. Immunohistochemically, the tumor cells are positive for CD56 (NCAM) and show luminal expression of EMA.

Differential diagnoses of ICCA with ductal plate malformation include von Meyenburg complex (biliary microhamartoma), biliary adenofibroma, and metastatic carcinoma. von Meyenburg complexes are small (usually 0.5 cm) and often located subcapsularly, and have irregularly shaped, angulated, or branching and dilated ducts, without atypia or mitotic activity [89]. Biliary adenofibroma is a solid-microcystic epithelial tumor consisting of microcystic and tubuloacinar glandular structures lined by a non-mucin-secreting biliary epithelium and supported by a fibrous stroma [90]. ICCAs with ductal plate malformation show more cytoarchitectural atypia and infiltrative growth than biliary adenofibromas. ICCAs with ductal plate malformation can mimic metastatic carcinomas. A clinical history of primary carcinoma at other sites and radiological findings are necessary to confirm the correct diagnosis.

### 9.3. Adenosquamous Carcinoma

Adenosquamous carcinoma is a subtype of ICCA that comprises significant proportions of both adenocarcinoma and squamous cell carcinoma [1,91]. It accounts for 2%–3% of ICCAs [3]. The extent of squamous differentiation required to designate a tumor as an adenosquamous carcinoma has not yet been defined. Some reported cases were associated with pre-existing conditions, such as hepatolithiasis and hepatic cysts lined by a stratified epithelium [92,93]. There are two theories regarding the tumorigenesis of intrahepatic adenosquamous carcinomas. One hypothesis is that the tumor develops from the metaplastic squamous epithelium lining the bile ducts, while the other possibility is the squamous differentiation of conventional adenocarcinoma. At diagnosis, 50% of patients have lymph node metastasis. The prognosis of adenosquamous carcinoma of the liver is worse than that of conventional ICCA [93].

Histologically, the tumor shows admixed components with squamous and glandular differentiation (Figure 5a,b). Squamous cell carcinoma components are irregularly intermixed with adenocarcinoma components. The degree of squamous cell differentiation ranges from well-formed keratin pearls to faint single-cell keratinization. Mucin production is present in the adenocarcinoma component. Tumors are larger than 5 cm in 60% of cases [91]. Necrosis and cystic degeneration are commonly present.

Differential diagnoses of adenosquamous carcinoma include mucoepidermoid carcinoma. Mucoepidermoid carcinoma and adenosquamous carcinoma show significant histological similarities and may pose challenges in differentiation. Adenosquamous carcinomas consist of distinct components of adenocarcinoma and squamous cell carcinoma, featuring well-developed keratinization and intercellular bridges but lacking intermediate cells. In contrast, mucoepidermoid carcinomas contain mixed mucous cells, and intermediate and epidermoid cells.

### 9.4. Squamous Cell Carcinoma

Pure squamous cell carcinoma arising as a primary tumor of the liver is extremely rare [94,95,96]. It has mostly been reported to be associated with hepatic cysts, hepatolithiasis, and hepatic teratoma [97,98,99]. The pathogenesis of squamous cell carcinoma is hypothesized to arise from the tumor transformation of the biliary epithelium in the presence of chronic inflammation, or the transformation of pre-existing liver cysts into metaplastic and then cancerous forms. However, its exact mechanism is still unknown [99]. The prognosis is very poor.

Histologically, the entire tumor shows squamous differentiation. Similar to squamous cell carcinomas at other sites, this tumor consists of cords, islands, or sheets of malignant squamous cells separated by a dense fibrous stroma. The extent of squamous cell differentiation varies between the anaplastic and mature keratinizing types. The tumor cells can be spindle-shaped and pleomorphic. Immunohistochemically, the tumor cells are positive for squamous cell markers (e.g., p40 and p63) and epithelial markers (e.g., CKs and EMA).

Differential diagnoses of squamous cell carcinoma include adenosquamous carcinoma, mucoepidermoid carcinoma, sarcomatoid carcinoma, and metastatic squamous cell carcinoma. Before diagnosing squamous cell carcinoma, extensive sampling should be performed to rule out the presence of glandular components in squamous cell carcinoma. Mucin staining may be performed to identify the foci of glandular differentiation and to exclude adenosquamous carcinoma with predominant squamous differentiation. Mucoepidermoid carcinomas must also be differentiated from squamous cell carcinomas based on the presence of mucus-secreting cells. Immunohistochemically, the presence of epithelial markers, such as keratin and epithelial membrane antigen, in the spindle cells is the best way to establish the squamous nature of the tumor. It is important to exclude metastatic squamous cell carcinoma or direct extension of carcinoma from the adjacent gallbladder.

### 9.5. Mucinous Carcinoma

Mucinous carcinoma is a type of invasive adenocarcinoma characterized by mucin-producing neoplastic cells floating in large amounts of extracellular mucus [100,101,102,103]. Focal mucinous features are sometimes present in conventional CCA; however, mucinous components are predominant in this subtype (>50% of the tumor). There is an association between the degree of mucin secretion and the presence and severity of clonorchiasis [104]. This may be an unusual complication of hepatolithiasis and recurrent pyogenic cholangitis [101]. Genomic analysis of mucinous ICCA revealed that the molecular carcinogenesis of mucinous ICCA differs from that of conventional ICCA [105,106]. It tends to have a more favorable prognosis than conventional ICCA [70].

Histologically, the tumor is composed of more than 50% extracellular mucin pools containing small nests of tumor cells. Most tumors have intestinal features such as scattered goblet cells and immunoreactivity to CDX2 and MUC2 [70]. The expression of the gastric mucin, MUC5AC, is also common, as in conventional CCA. Non-mucinous components are occasionally papillary adenocarcinomas. BilIN may be present in the adjacent bile duct, particularly in patients with PSC, suggesting a dysplasia–carcinoma sequence [107]. IPNB can also progress to mucinous carcinoma if it becomes invasive, but is currently categorized separately (IPNB with an associated mucinous carcinoma) [108,109].

Differential diagnoses of mucinous carcinoma include mucus lakes, conventional ICCA, and metastatic mucinous carcinoma originating in other organs, such as the gastrointestinal tract, pancreas, breast, and ovary. Mucus lakes are characterized by the accumulation of mucin within the liver tissue and are commonly observed in association with certain tumors that produce excessive mucin. Mucus lakes do not contain floating cancer cells. While the presence of a mucus lake is not indicative of cancer itself, it is a feature often seen in association with cancer. Conventional ICCAs may have small mucinous areas; however, tumors are not classified as the mucinous type unless more than 50% of the tumor is mucinous. A previous history of mucinous carcinomas of other organs raises the suspicion of metastasis.

### 9.6. Signet Ring Cell Carcinoma

Signet ring cell carcinoma (SRCC) is a rare subtype of CCA with predominantly or exclusively signet ring cells (>50% of the tumor cells), similar to signet ring cell carcinomas at other sites. SRCCs arising from the intrahepatic bile duct are extremely rare [110,111], and SRCCs originating from the hilar and distal bile ducts have also been reported [112,113,114]. The prognosis of patients with SRCCs arising in the gastrointestinal tract is generally worse than that of patients with adenocarcinoma, NOS, and mucinous carcinomas. However, the prognosis of SRCC arising from the intrahepatic bile duct remains unclear.

Histologically, the tumor predominantly consists of incohesive signet ring cells. Signet ring cells are characterized by clear, rounded droplets of cytoplasmic mucin with an eccentrically placed nucleus. The degree of nuclear atypia in the signet ring cells varies from mild to marked. The signet ring cells may form a lace-like glandular or delicate microtrabecular pattern. Signet ring cells can be observed in the pools of extracellular mucin. In some signet ring cell carcinomas, foci of undifferentiated cells and well-differentiated neoplastic glands may also be observed. Neoplastic signet ring cells are positive for Mayer’s mucicarmine, Alcian blue, and PAS staining. Immunohistochemically, the tumor cells are usually positive for CK7, CEA, MUC2, and MUC5AC and rarely positive for CDX2.

Differential diagnoses of SRCC include conventional ICCA and mucinous carcinoma. If a tumor consists predominantly of well-differentiated neoplastic glands and contains isolated signet ring cells, or if < 50% of the tumor consists of signet ring cells, it should be classified as conventional ICCA with a signet ring cell component. Because some signet ring cell carcinomas contain abundant extracellular mucin, they are often confused with mucinous carcinomas. It should be kept in mind that signet ring cells are not the predominant component of mucinous carcinomas. Histiocytes that phagocytose mucin may be mistaken for signet ring cells. On immunohistochemical staining, these mucinophages are negative for CK and CEA.

### 9.7. Clear Cell Cholangiocarcinoma

Clear cell cholangiocarcinoma (CCA) is a rare ICCA subtype characterized by extensive clear cell change [115,116,117]. Focal clear cell changes are common in ICCA; however, extensive or nearly complete clear cell changes are very rare in ICCA. The underlying factors and risk elements associated with clear cell CCA remain unclear. Clear cell changes have been attributed to the presence of intracytoplasmic glycogen, mucin, or lipids based on histochemistry and electron microscopy [117,118,119]. This tumor may arise from reactive bile ducts or cholangiomatous lesions [115]. Its prognosis is better than that of conventional ICCA [118].

Histologically, the tumor comprises cuboidal, columnar, or polygonal cells with clear cytoplasm. The tumor cells are arranged in variable proportions in tubular structures, solid sheets, cords, trabeculae, and papillary structures. A desmoplastic stroma is present. Immunohistochemically, the tumor cells exhibit positive CD56 (NCAM) expression and are negative for S100 protein and vimentin [115,120].

Differential diagnoses of clear cell CCA include atypical bile duct adenoma, clear cell type, clear cell HCC, and metastatic clear cell carcinomas of the kidney, gastrointestinal tract, lung, and thyroid gland [115]. Compared with the atypical clear cell type of bile duct adenoma, clear cell CCAs exhibit more significant nuclear atypia and increased mitotic activity [121]. Clear cell HCCs are positive for markers of hepatocytic differentiation, such as Hep Par-1, arginase-1, CD10 (canalicular staining), polyclonal CEA (canalicular staining), and α-fetoprotein. Metastatic clear cell renal cell carcinomas are positive for PAX8, carbonic anhydrase IX (CAIX), and renal cell carcinoma marker (RCC-Ma). Metastatic clear cell carcinomas of the lung and thyroid gland are positive for TTF1.

### 9.8. Mucoepidermoid Carcinoma

Mucoepidermoid carcinoma of the liver is a rare subtype of ICCA. Similar to the prototype in the salivary gland, it is defined as a malignant epithelial neoplasm composed of mucous, intermediate, and epidermoid (squamoid) cells [45,70,122]. Grossly, the tumors do not differ from conventional ICCA, particularly of the mass-forming type. A potential origin of this variant is the peribiliary glands. *CRTC1*::*MAML2* fusion is a distinct molecular alteration identified in mucoepidermoid carcinomas of the salivary glands [123,124]. However, most cases of hepatic mucoepidermoid ICCA lack *CRTC1*::*MAML2* fusion, and only one patient with hepatic mucoepidermoid ICCA harboring a *CRTC1*::*MAML2* fusion has been reported [125]. Recently, a case of hepatic mucoepidermoid carcinoma associated with germline mutations in Fanconi’s anemia gene and somatic mutations in *GNAS* R201H was reported [126]. This tumor appears to be an aggressive tumor with a poor prognosis [122,127].

Histologically, the three cellular components, i.e., mucous, intermediate, and epidermoid cells, are easily identifiable, but their proportions vary from case to case. The tumors are predominantly solid, but small cystic spaces lined by neoplastic cells may also be present. Intermediate and epidermoid cells may exhibit clear cytoplasm. The grading scheme used for mucoepidermoid carcinoma of the salivary glands is not applicable to intrahepatic tumors because the liver equivalent is always of a high grade. Immunohistochemically, the epidermoid cells are positive for p40 and p63 [128].

Differential diagnoses of mucoepidermoid carcinoma include conventional ICCA, adenosquamous carcinoma, and squamous cell carcinoma. Severe nuclear atypia, numerous mitoses, widespread necrosis, and extensive keratinization favor the possibility of poorly differentiated conventional ICCA or adenosquamous carcinoma rather than mucoepidermoid carcinoma. Features that favor the diagnosis of mucoepidermoid carcinoma over squamous cell carcinoma include the identification of mucous cells and the presence of low-grade mucoepidermoid carcinoma components. A diagnosis of high-grade mucoepidermoid carcinoma should only be made if typical morphologic features are present, or if a recognizable component of low-grade mucoepidermoid carcinoma can be identified.

### 9.9. Lymphoepithelioma-like Cholangiocarcinoma

Lymphoepithelioma-like cholangiocarcinoma (LELCCA) is a rare subtype of ICCA characterized by an associated prominent, non-neoplastic lymphoplasmacytic cell infiltrate [129,130]. To date, more than 60 cases have been reported in the literature [131,132,133,134,135,136,137]. These tumors are strongly associated with Epstein–Barr virus (EBV) [128]. Most cases are diagnosed in East Asia [9]. Frequent *pTERT* and *TP53* mutations are detected [136]. Approximately 50% of LELCCs express programmed death-ligand 1 (PD-L1) [136]. High PDL-1 expression in LELCCA has implications for potential immunotherapeutic strategies. This subtype has a favorable prognosis compared with conventional ICCA [133].

Histologically, the tumor is composed of an undifferentiated or poorly differentiated form of adenocarcinoma with dense lymphoplasmacytic infiltration. The tumor cells are large and polygonal, with vesicular nuclei and prominent nucleoli. The cells are arranged in solid sheets, nests, cords, or tubules with indistinct intercellular borders. The lymphoid stroma consists of a mixture of T cells, B cells, and mature plasma cells. Neutrophils and eosinophils are rare. In a subset of cases, the tumor cells are positive for EBV-encoded small RNAs (EBER) in in situ hybridization [131]. Immunohistochemically, the tumor cells are positive for biliary-type CKs (CK7 and CK19) and stemness markers (CD133 and EpCAM) [133].

Differential diagnoses of LELCCA include conventional ICCA, lymphocyte-rich HCC, and metastatic lymphoepithelioma-like carcinoma of other organs, such as the nasopharynx. Conventional ICCAs may show lymphoplasmacytic infiltration, whereas the lymphoid stroma may be less prominent in lymphoepithelioma-like CCAs. Numerous intraepithelial lymphocytes can provide a diagnostic clue in equivocal cases. The confirmation from EBER positivity is helpful in the diagnosis of LELCCA. Lymphoepithelioma-like HCCs and metastatic carcinomas can also have a lymphocyte-rich morphology. Therefore, to identify cholangiocytic differentiation, it is necessary to conduct immunohistochemical staining. Lymphoepithelioma-like HCCs are positive for markers associated with hepatocyte differentiation, such as Hep Par-1 and arginase 1.

### 9.10. Sarcomatous Cholangiocarcinoma

Sarcomatous cholangiocarcinoma (CCA) (also known as sarcomatoid CCA or spindle cell carcinoma) is a rare subtype of ICCA predominantly composed of neoplastic spindle or pleomorphic giant cells [138,139]. Its pathogenesis remains unclear. Two hypotheses have been proposed [135,136,137,138]. The first is the sarcomatous differentiation of primary carcinoma cells of epithelial origin, explained by the process of epithelial–mesenchymal transition (EMT). The second hypothesis involves the biphasic differentiation of pluripotent stem cells, which can develop into either carcinomas or sarcomas. Chronic hepatitis B and C, cirrhosis, and hepatolithiasis are often observed in the surrounding liver [140,141,142,143,144]. Mutations in *TP53* and in the promoter of *TERT* (*pTERT*) have rarely been identified [141]. This subtype generally shows aggressive clinical behavior and confers a poorer prognosis than ordinary ICCA.

Histologically, the tumor cells have spindle-shaped or pleomorphic giant nuclei and are arranged in sheets or fascicular patterns. Osteoclast-like giant cells can also be found [142]. Mitotic figures are frequently present. Necrosis is common. Adenocarcinoma components with glandular morphology are often focally observed. Immunohistochemically, the sarcomatous components at least focally express epithelial markers such as CKs and EMA, confirming the epithelial nature. Mesenchymal markers, such as vimentin, are frequently positive in sarcomatous ICCA [145,146].

Differential diagnoses of sarcomatous CCA include sarcomatoid HCC, carcinosarcoma, and spindle cell sarcomas. Sarcomatoid HCCs are focally positive for markers of hepatocyte differentiation and lack specific mesenchymal differentiation. Foci of moderately to poorly differentiated HCC components may also be present. Sarcomatoid CCAs should be distinguished from carcinosarcoma. The diagnostic term carcinosarcoma should be restricted to tumors with proven heterologous sarcomatous elements such as osteosarcoma, chondrosarcoma, and rhabdomyosarcoma. Sarcomatous ICCA can mimic any spindle cell sarcoma. The presence of an adenocarcinoma component is helpful in the diagnosis. The diagnosis of sarcomas must be confirmed through immunohistochemical studies. For example, metastatic gastrointestinal stromal tumors are positive for CD117 and DOG1.

Table 3 summarizes the clinical, pathological, and molecular characteristics of ICCA subtypes as recognized in the 2019 WHO classification of liver tumors.

## 10. New Provisional Subtypes of Intrahepatic Cholangiocarcinoma Not Included in the 2019 WHO Classification

The following sections detail the new provisional subtypes of ICCA that were not incorporated into the 2019 WHO classification of liver tumors. These proposal subtypes are mainly based on their unique morphological appearance. These subtypes are classified as provisional owing to the limited published data.

### 10.1. Tubulocystic Carcinoma of the Bile Duct

Tubulocystic carcinoma of the bile duct (TCCBD) has recently been suggested [147]. It is morphologically characterized by a distinct tubulocystic pattern, very similar to that of tubulocystic carcinoma of the kidney [148]. In a series of studies involving six cases, three cases developed in the liver and three cases developed in the perihilar region [147]. The mean age was 66 years (range, 44 to 78 years), and the mean tumor size was 4.4 cm (range, 3 cm to 7 cm). This tumor type represents a peculiar, indolent form of invasive carcinoma [147].

Histologically, the tumors are characterized by cystically dilated tubules of varying sizes, presenting an overall spongy or honeycomb-like appearance [45]. The lining epithelium consists of cuboidal to flat cells with mild atypia. There is no intracytoplasmic mucin. Cytoplasmic projections resembling apocrine snouts are often present. Eosinophilic intraluminal contents are frequently present. Intracystic papillary or tubular proliferation is observed. In a case study of TCCBD, the tumor cells were immunohistochemically positive for CD10, CAM5.2, and vimentin [149].

Differential diagnoses of TCCBD include bile duct hamartoma, biliary adenofibroma, and mucinous cystic neoplasm. Bile duct hamartomas and biliary adenofibromas can mimic TCCBD. The presence of infiltration into the perineural space or the extrahepatic connective tissue and intracystic papillary growth favor TCCBD. In contrast to mucinous cystic neoplasms of the liver, TCCBDs do not have an ovarian-like stroma.

### 10.2. Cholangioblastic Cholangiocarcinoma

Cholangioblastic CCA (also called solid tubulocystic carcinoma or thyroid follicle-like carcinoma) is a recently described subtype of ICCA [150,151,152,153,154,155,156]. It tends to affect younger women (with an average age of approximately 40 years). There is no underlying chronic liver disease [157]. This subtype is usually large and solitary and shows a multinodular appearance. Recently, *NIPBL*::*NACC1* fusion was identified that appeared to be characteristic of the cholangioblastic subtype [150,157]. Tumor recurrence occurs in approximately 60% of cases. The prognosis remains unclear.

Histologically, the tumor cells have round nuclei, evenly dispersed chromatin, and eosinophilic cytoplasm. The tumors exhibit a wide range of morphologies, including solid, trabecular, microcystic, follicular, and blastema-like areas [156]. The follicle-like structures may contain eosinophilic pink secretions that mimic thyroid follicles. Mitotic activity varies and is typically low (~10/10 high-power fields). The coexistence of smaller cells with a scant cytoplasm can result in a biphasic cytological appearance [150]. Immunohistochemically, the tumor cells are positive for CK7 and CK19, along with the patchy expression of chromogranin and synaptophysin. Diffuse and strong immunoreactivity to inhibin alpha is a characteristic finding [157].

Differential diagnoses of cholangioblastic CCA include HCC and neuroendocrine tumors. Solid and trabecular areas may resemble HCC and neuroendocrine tumors. HCCs are positive for markers of hepatocellular differentiation (e.g., Hep Par-1 and arginase-1). Cholangioblastic CCA subtypes are negative for hepatocellular markers. These subtypes show a patchy expression of chromogranin and synaptophysin but do not express insulinoma-associated protein 1 (INSM1). In contrast, well-differentiated neuroendocrine tumors typically exhibit a diffuse expression of neuroendocrine markers [150].

### 10.3. Enteroblastic Cholangiocarcinomae

Enteroblastic differentiation has been reported in carcinomas of various organs [158,159,160,161]. It is the most common in gastric cancer and is called hepatoid carcinoma, alpha-fetoprotein (AFP)-producing carcinoma, or adenocarcinoma with enteroblastic differentiation based on the predominant microscopic features [162,163]. Carcinomas with hepatoid or enteroblastic differentiation are extremely rare in the liver and extrahepatic bile ducts (EHBDs) [164,165]. Recently, a molecular study on hepatoid tumors of various organs showed that biliary hepatoid tumors commonly display a loss of *CDKN2A* and loss of chromosome 18 [166]. Hepatoid adenocarcinoma of the stomach generally exhibits more aggressive behavior and a worse prognosis than conventional gastric cancer [167]. Hepatoid carcinoma and related entities in EHBD may be more aggressive than conventional extrahepatic CCA [165].

Histologically, the tumor mainly comprises cuboidal or columnar cells with clear cytoplasm resembling the fetal gut epithelium. Part of the tumor may have polygonal-shaped cells with eosinophilic cytoplasm and tubular, papillary, trabecular, and solid patterns, showing characteristics of hepatoid differentiation. In some areas, the tumor cells can show high-grade nuclear atypia and mitoses. The mucus is inconspicuous. Immunohistochemically, the tumor cells are positive for Hep Par-1, arginase-1, glypican-3, and AFP to varying degrees. SALL4 is a good sensitive marker for enteroblastic differentiation [168].

Differential diagnoses of enteroblastic CCA include HCC, combined hepatocellular cholangiocarcinoma (cHCC-CCA), and metastatic carcinoma. It is difficult to distinguish enteroblastic CCA from HCC and cHCC-CCA. SALL4 is helpful for differential diagnosis. However, SALL4 expression should be interpreted cautiously because SALL4 can be focally positive in HCC [169]. If adenocarcinomas exhibit a solid pattern or contain clear cells, the possibility of enteroblastic CCA should be of particular concern. If hepatoid carcinoma is observed in the liver, the possibility of metastatic cancer, especially from the stomach, should be meticulously ruled out [170].

### 10.4. Micropapillary Carcinoma

Micropapillary carcinoma is a distinct histological subtype of carcinoma that can occur in various organs [171,172]. This histological subtype was first described in breast cancer and is characterized by delicate filiform processes and infiltrating clusters of micropapillary aggregates without fibrovascular cores [173]. Micropapillary carcinomas of the ampullo-pancreato-biliary region have rarely been reported [174,175]. Similar to other organs, the micropapillary pattern is associated with a highly aggressive behavior, frequent lymph node metastasis, and a short survival period (a median survival period of 8 months) [175]. Micropapillary carcinoma of the ampullo-pancreato-biliary region is a poorly differentiated (high-grade) subtype of adenocarcinoma [175].

Histologically, the tumor shows small, closely packed micropapillary clusters (without fibrovascular cores) surrounded by clear spaces due to a stromal retraction artifact. The tumor cells are columnar or cuboidal and exhibit eosinophilic cytoplasm and a moderate degree of nuclear atypia. Abundant neutrophil infiltration is commonly observed. Lymphovascular invasion is frequent. Immunohistochemically, the tumor cells are positive for galectin-3 and E-cadherin, molecules that are implicated in abnormalities of tumor cell–stroma adhesion [175].

Differential diagnoses of micropapillary carcinoma include conventional ICCA, mucinous ICCA, and metastatic micropapillary carcinoma. Conventional ICCAs are distinguished from micropapillary carcinomas by the absence of a true micropapillary architecture, which lacks fibrovascular cores. Mucinous ICCAs have large extracellular mucin pools, but micropapillary carcinomas do not have large extracellular mucin pools. Correlation between imaging findings and patient history can aid in the diagnosis of metastatic micropapillary carcinoma.

### 10.5. Acinar Cell Carcinoma

Acinar cell carcinoma is a malignant epithelial neoplasm characterized by acinar cell differentiation. Primary acinar cell carcinomas of the liver have rarely been reported in the literature [176,177,178]. The tumors are typically large (mean size, 12 cm) and form solid masses. Neoplasms exhibiting acinar cell differentiation have rarely been reported beyond the pancreas, especially in the gastrointestinal tract and ampulla of Vater [179,180,181]. The origin and development of hepatic acinar cell carcinoma remain unclear. The prognosis appears favorable [70,176,177].

Histologically, the tumor cells have uniform, basally located round nuclei and possess a moderate amount of granular, eosinophilic to amphophilic cytoplasm, which includes zymogen granules. The tumor cells are predominantly arranged in acinar, glandular, trabecular, or solid nest patterns. Immunohistochemically, the tumor is positive for acinar cell differentiation markers, such as trypsin, chymotrypsin, amylase, and lipase. Synaptophysin and chromogranin may be focally positive.

Differential diagnoses of acinar cell carcinoma include HCC, neuroendocrine tumor, and metastatic acinar cell carcinoma of the pancreas and other organs. HCCs are positive for markers of hepatocellular differentiation, such as Hep Par-1, arginase-1, polyclonal CEA (canalicular staining), CD10 (canalicular staining), and AFP. Neuroendocrine tumors have salt and pepper chromatin and are positive for synaptophysin, chromogranin, and INSM1. It is essential to exclude metastases from clinically occult acinar cell carcinomas of the pancreas and other organs [70]. Metastatic acinar cell carcinoma usually presents as multiple liver masses. Imaging diagnostics, examinations of other organs, and clinical history are crucial for the differential diagnosis.

Table 4 summarizes an overview of the clinical, pathological, and molecular features of the newly proposed subtypes of ICCA, which were not part of the 2019 WHO classification.

## 11. Pathological Diagnostic Approach

### 11.1. Specimen Handling

#### 11.1.1. Biopsy Specimens

Core needle biopsies are commonly used to evaluate liver diseases that affect liver function (e.g., chronic hepatitis) and are also used to evaluate liver masses (e.g., primary liver tumors). For a biopsy sample, the length and diameter should be measured, and the color should be described. The sample should undergo routine histological examination unless additional testing is required. Hematoxylin and eosin (H&E) staining is a standard technique used to diagnose liver disorders. Step sections are favored compared with serial sections, making the intervening sections available for histochemical and IHC staining [182]. For molecular research or electron microscopy examination, fresh tumor specimens can be utilized [183].

#### 11.1.2. Hepatectomy Specimens

After identifying the procedure performed and orienting the specimen, pathologists weigh and measure the hepatectomy specimen. The step may necessitate assistance from a surgeon [179]. Margins should be identified and inked. The liver is serially sectioned perpendicularly to the resection margin at thin intervals (<1 cm), and all cut surfaces for liver mass lesions are thoroughly examined [182]. Pathologists should evaluate the number and size of liver tumors, color, consistency, necrosis, distance between the tumor and closest surgical margin, and any macroscopic alterations in the non-cancerous liver tissue (e.g., cirrhosis). It is advised that a minimum of one tissue block for every centimeter of the tumor’s area is collected in the case of larger tumors. Samples should be taken from all distinct tumor areas and the transitional zones between other regions [184].

### 11.2. Pathological Diagnostic Approach

For an accurate diagnosis of ICCA, obtaining a biopsy sample representative of the lesion and ensuring proper tissue processing are necessary. A systemic diagnostic approach achieves the precise diagnosis of liver mass lesions [185]. The first and most critical step in the diagnostic process is a comprehensive histological evaluation of sections stained with H&E at a low magnification. Pathologists should initially confirm the presence of tissue with lesions and carefully analyze tumor cell morphology, architectural growth patterns, and stromal characteristics. Clinical history, imaging results, and cancer serum markers (e.g., CA19-9) can aid in diagnosing ICCA. IHC and molecular studies are useful in diagnosing challenging cases and detecting uncommon variants of ICCA. The diagnostic algorithm used for the ICCA is illustrated in Figure 6.

ICCAs should be differentiated from benign biliary lesions (e.g., ductular reaction, peribiliary gland hyperplasia, bile duct adenoma, and biliary adenofibroma), HCC, combined HCC-CCA, epithelioid hemangioendothelioma, and metastatic carcinoma, especially from the pancreas, gallbladder, and extrahepatic bile ducts [186,187,188]. Distinguishing between ICCA and metastatic carcinoma can be challenging because of the similarities in clinical presentation, overlapping histologic features, and the lack of ICCA-specific tissue markers.

The immunohistochemical markers commonly used to differentiate ICCA from metastatic carcinoma are listed in Table 5.

## 12. Future Perspectives

The current classification of ICCA based on anatomical origin poses several problems, particularly in accurately assessing the epidemiological background, carcinogenesis, and patient outcome [229]. ICCA is highly heterogeneous in terms of cell origin, tissue structure, immunophenotype, and molecular mutations. The criteria for the histological classification of ICCA, the selection of immunohistochemical markers, the detection of molecular targets, and the differential diagnosis of histological subtypes need to be refined for better applications in practice [230].

Recently, ICCAs have been divided into four distinct immune subclasses, each associated with varying prognostic outcomes [231]. The tumor immune microenvironment (TIME) is a dynamic and complex system formed by the interactions of tumor cells with mesenchymal cells, including various immune cells, endothelial cells, and fibroblasts, as well as a range of cytokines [232]. In ICCA, the TIME plays a crucial role in tumor growth, invasion, angiogenesis, and metastasis through its interactions with tumor cells [233]. Recent studies have identified novel TIME-based subtypes of CCA, each characterized by distinct mechanisms of immune escape and patient outcomes [234]. However, the immunobiology, antitumor immunity, and immunotherapy for ICCAs are still poorly understood. Further studies in these fields are required.

Next-generation sequencing (NGS) has improved our understanding of the tumor biology of ICCA by uncovering the complex and diverse genomic landscape of this disease [235]. Despite these advancements, several important issues still need to be resolved, such as improving the identification of risk factors linked to this disease and an understanding of their role in the genetic heterogeneity of ICCA. For advancements in managing ICCA in the near future, close collaboration between basic science and clinical research is essential [236].

## 13. Conclusions

ICCAs are a group of genetically, pathologically, and clinically heterogeneous tumors. Their incidence has been increasing worldwide. Recent advances in the molecular pathology and histological subtypes of ICCA have significantly improved our understanding of its biology. An accurate diagnosis of ICCA requires a combination of clinical and radiological findings, histological findings, and immunohistochemical and molecular analyses. ICCA remains a disease associated with a significant risk of mortality. Further studies are required to deepen our understanding of its biology and identify novel biomarkers and potential therapeutic targets.

## Figures and Tables

**Figure 1 cancers-16-01537-f001:**
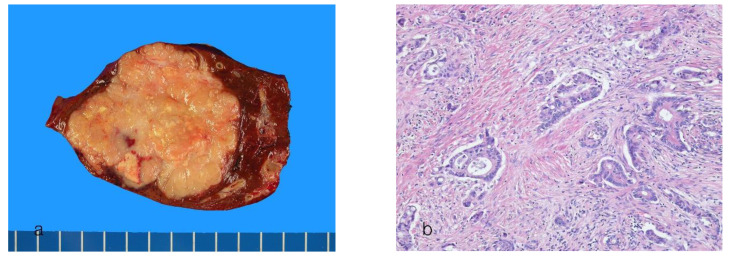
Intrahepatic cholangiocarcinoma, small-duct type. (**a**) A well-defined, lobulated, yellow-gray mass is present. (**b**) The tumor cells are arranged in a glandular pattern in fibrous stroma (hematoxylin-eosin stain, ×100).

**Figure 2 cancers-16-01537-f002:**
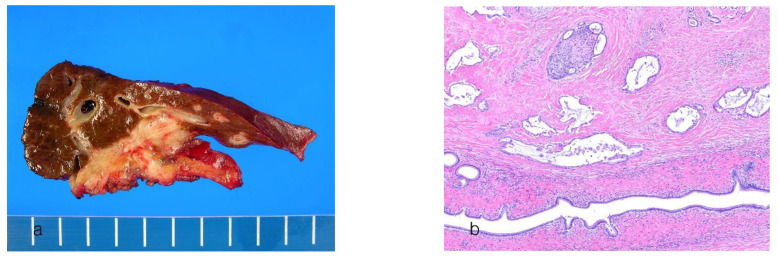
Intrahepatic cholangiocarcinoma, large-duct type. (**a**) A gray-white, periductal-infiltrating tumor. (**b**) The infiltrating dilated tumor glands along the large bile duct wall. Desmoplastic stroma and perineural invasion are present (hematoxylin–eosin stain, ×40).

**Figure 3 cancers-16-01537-f003:**
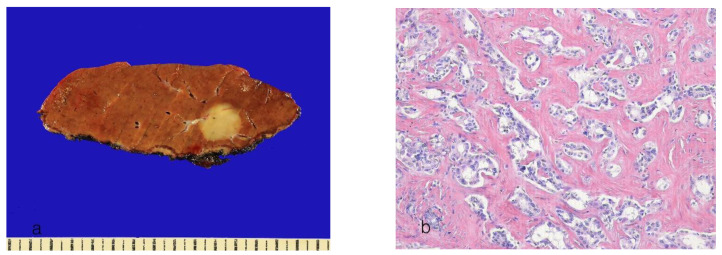
Intrahepatic cholangiocarcinoma, cholangiolocarcinoma subtype. (**a**) A well-defined, gray-yellow, solid tumor. (**b**) Tumor cells arranged in a small glandular and cord-like pattern in hyaline fibrous stroma (hematoxylin–eosin stain, ×100).

**Figure 4 cancers-16-01537-f004:**
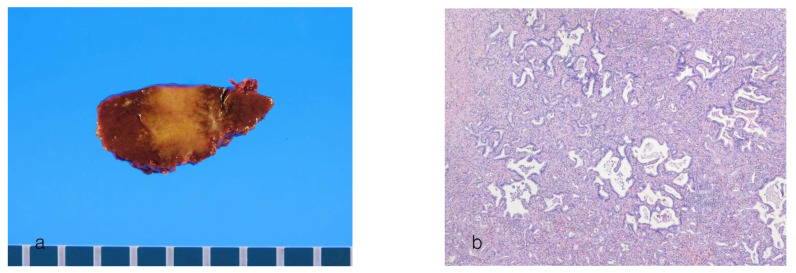
Intrahepatic cholangiocarcinoma with a ductal plate malformation pattern. (**a**) A gray-white nodular tumor with irregular border. (**b**) The tumor showing a ductal plate malformation pattern with irregularly dilated lumens (hematoxylin–eosin stain, ×40).

**Figure 5 cancers-16-01537-f005:**
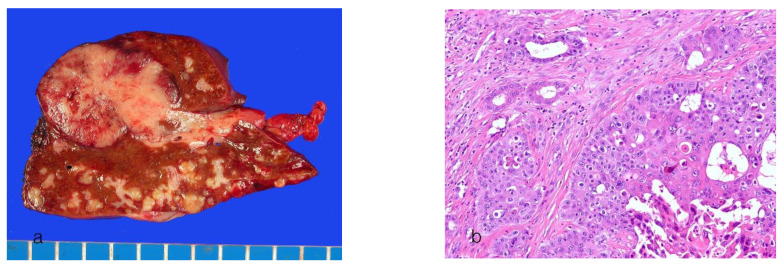
Intrahepatic cholangiocarcinoma, adenosquamous carcinoma subtype. (**a**) A well-defined, white-pink, solid tumor. (**b**) Adenocarcinoma components with tubule formation intermixed with squamous cell carcinoma components with keratinization (hematoxylin–eosin stain, ×100).

**Figure 6 cancers-16-01537-f006:**
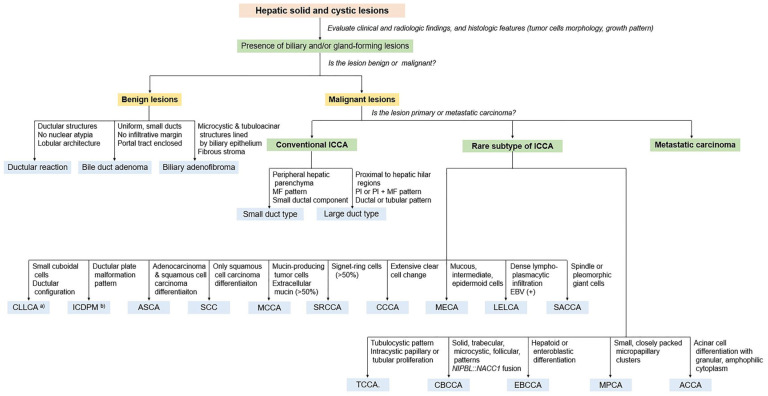
Pathological diagnostic approach of intrahepatic cholangiocarcinoma based on clinical and radiologic findings, histologic features (tumor cells morphology, growth pattern), and immunohistochemistry. ICCA, intrahepatic cholangiocarcinoma; CLLCA, cholangiolocarcinoma; ICDPM, intrahepatic cholangiocarcinoma with ductal plate malformation pattern; ASCA, adenosquamous carcinoma; SCC, squamous cell carcinoma; MCCA, mucinous carcinoma; SRCCA, signet-ring cell carcinoma; CCCA, clear cell carcinoma; MECA, mucoepidermoid carcinoma; LELCA, lymphoepithelioma-like carcinoma; SACCA, sarcomatous cholangiocarcinoma; TCCA, transitional cell carcinoma; CBCCA, cholangioblastic cholangiocarcinoma; EBCCA, enteroblastic cholangiocarcinoma; MPCA, micropapillary carcinoma; ACCA, acinar cell carcinoma. ^(a)^ CLLCA and ^(b)^ ICDPM are considered small-duct type of conventional ICCA.

**Table 1 cancers-16-01537-t001:** Evolution of the World Health Organization (WHO) classification of intrahepatic cholangiocarcinoma.

	2000 WHO Classification (3rd Edition)	2010 WHO Classification (4th Edition)	2019 WHO Classification (5th Edition)
Tumor Category	Epithelial tumors, malignant	Epithelial tumors: biliary, malignant	Malignant biliary tumors
Tumor type and subtypes	Intrahepatic CCA (peripheral bile duct carcinoma)	Intrahepatic CCA	Conventional intrahepatic CCA Large duct intrahepatic CCA Small duct intrahepatic CCA
			Cholangiolocarcinoma Intrahepatic CCA with ductal plate malformation pattern
	Subtypes	Subtypes	Subtypes
	Cholangiolocellular carcinoma	Combined HCC-CCA with stem cell features, cholangiolocellular type ^(a)^	
	Adenosquamous carcinoma	Adenosquamous carcinoma	Adenosquamous carcinoma
	Squamous cell carcinoma	Squamous cell carcinoma	Squamous cell carcinoma
	Mucinous carcinoma	Mucinous carcinoma	Mucinous carcinoma
	Signet-ring cell carcinoma	Signet-ring cell carcinoma	Signet-ring cell carcinoma
	Clear cell carcinoma	Clear cell carcinoma	Clear cell carcinoma
	Mucoepidermoid carcinoma	Mucoepidermoid carcinoma	Mucoepidermoid carcinoma
	Lymphoepithelioma-like carcinoma	Lymphoepithelioma-like carcinoma	Lymphoepithelioma-like carcinoma
	Sarcomatous intrahepatic CCA	Sarcomatous intrahepatic CCA	Sarcomatous intrahepatic CCA

^(a)^ Classified as a malignancy of mixed or uncertain origin according to the 2010 WHO classification (4th edition) of liver tumors. CCA, cholangiocarcinoma; combined HCC-CCA, combined hepatocellular–cholangiocarcinoma.

**Table 2 cancers-16-01537-t002:** Characteristics of small-duct type and large-duct type conventional intrahepatic cholangiocarcinoma ^(a)^.

	Small-Duct Type	Large-Duct Type
Main location	Peripheral hepatic parenchyma	Proximal to hepatic hilar regions
Risk factors	Hepatitis virus (HBV and HCV infection), alcoholic liver disease, metabolic syndrome, hemochromatosis, diabetes mellitus, obesity	Primary sclerosing cholangitis, hepatolithiasis, liver fluke infection
Precursors	Unknown	BilIN, IPNB, ITPN
Goss features	MF pattern	PI pattern, PI + MF pattern
Origins of cells	Small bile ducts and bile ductules, hepatic progenitor cells?	Intrahepatic large bile ducts, peribiliary glands
Histology	Small-ductal components: tubular pattern with low columnar to cuboidal cells and desmoplastic reaction Ductular components, cuboidal epithelia showing ductular or cord-like pattern with slit-like lumen and desmoplastic reaction	Ductal or tubular pattern with columnar to cuboidal epithelium, with desmoplastic reaction
Mucin production	Non–mucin-secreting glands	Mucin-secreting glands
Perineural/lymphatic invasion	Can be present	Common
Tumor border	Expansile or pushing, rarely infiltrative	Infiltrative
Molecular features	*BAP1, IDH1/2* mutations, *FGFR2* fusions, *SMAD4, BAP1, BRAF, ARIDA1A, KRAS, TP53, SMAD4* mutations	*KRAS* mutations, *TP53* mutations, *SMAD4* mutations, *MDM2* amplification
Immunohistochemical features		
Common markers	EMA (MUC1), cytokeratin 7, cytokeratin 19	EMA (MUC1), cytokeratin 7, cytokeratin 19
Characteristic markers	CD56 (NCAM), C-reactive protein, N-cadherin, BAP1 (loss)	MUC5AC, MUC6, S100P, TFF1, AGR2, MMP7, SMAD4 (loss)
Similar to	Adenocarcinoma component of combined HCC-CCA	Perihilar CCA
Prognosis	Favorable (5-year survival 35–40%)	Poor (5-year survival 20–25%)

^(a)^ Data are based on the 2019 WHO classification (5th edition) of liver tumors [1]. BilIN, Biliary intraepithelial neoplasia; IPNB, intraductal papillary neoplasm of the bile ducts; ITPN; intraductal tubulopapillary neoplasm of the bile ducts; MF, mass-forming; PI, periductal infiltrating; combined HCC-CCA, combined hepatocellular-cholangiocarcinoma; BAP1, BRCA1; BRCA1 Associated Protein-1; TFF1, trefoil factor 1; ARG2, anterior gradient 2; MMP7, matrix metalloproteinase 7; SMAD4, SMAD family member 4; NCAM, neural cell adhesion molecule.

**Table 3 cancers-16-01537-t003:** Clinical, pathological, and molecular features of the rare subtypes of intrahepatic cholangiocarcinoma ^(a)^.

Subtype	Relative Frequency	Clinical Features	Pathological Features	Molecular Features	Prognosis ^(b)^	References
Cholangiolocarcinoma	<5%	Share clinical and imaging features with both HCC and ICCA	>80% of the tumor composed of a ductular configuration, small cuboidal cells with round to oval nuclei with fine chromatin and scant cytoplasm, hyalinized fibrotic stroma	No distinct findings to date	Better	[82,83,84]
ICCA with ductal plate malformation pattern	<5%	60% of patients have a history of chronic liver disease	>50% of the tumor shows tumor structures resembling those of ductal plate malformation	*FGFR2* and *PTPRT* are most frequently mutated	Similar	[70,87,88]
Adenosquamous carcinoma	2–3%	Associated with hepatolithiasis, hepatic cysts	Tumor composed of both glandular and squamous cell differentiation	No distinct findings to date	Worse	[91,92,93]
Squamous cell carcinoma	<1%	Associated with hepatic cyst, hepatolithiasis, hepatic teratoma	Entire tumor shows squamous differentiation	No distinct findings to date	Worse	[94,95,96,97,98,99,100,101]
Mucinous carcinoma	<1%	Unusual complication of hepatolithiasis and recurrent pyogenic cholangitis	>50% of the tumor composed of extracellular mucin pools and clusters of tumor cells; BilIN and IPNB can progress to mucinous carcinoma	Mucin synthesis by MUC4 and MUC16 is elevated via the up-regulated expression of mesothelin; transcription factor ONECUT3	Better	[100,101,102,103,104,105,106]
Signet ring cell carcinoma	<1%	No distinct findings to date	>50% of the tumor composed of signet ring cells	No distinct findings to date	Unclear	[110,111,112,113,114]
Clear cell carcinoma	<1%	No distinct findings to date	>50% of the tumor composed of clear cells	No distinct findings to date	Better	[115,116,117,120]
Mucoepidermoid carcinoma	<1%	No distinct findings to date	Tumor composed of mucous, intermediate, and epidermoid cells	Most cases lack *CRTC1*::*MAML2* fusion	Worse	[122,123,124,125,126]
Lymphoepithelioma-like carcinoma	<5%	Associated with Epstein–Barr virus	Tumor cells arranged in sheets and abortive glands with dense lymphoplasmacytic infiltrate	Frequent mutations in *pTERT* and *TP53*	Better	[129,130,131,132,133,134,135,136,137]
Sarcomatous ICCA	<5%	Associated with chronic hepatitis B and C, hepatolithiasis	Spindle or pleomorphic giant tumor cells with focal adenocarcinoma component	Rare mutations in *pTERT* and *TP53* mutations	Worse	[138,139,140,141,142,143,144]

^(a)^ The ICCA subtypes are based on the 2019 WHO classification (5th edition) of liver tumors [1], ^(b)^ compared with the prognosis of conventional intrahepatic cholangiocarcinoma. ICCA, intrahepatic cholangiocarcinoma; IL-6, interleukin-6; JAK, Janus kinase; STAT, signal transducer and activator of transcription; *TERT*, telomerase reverse transcriptase; *FGF19*, fibroblast growth factor 19; *TSC1/2*, tuberous sclerosis complex 1/2; TGF-β, transforming growth factor-β; G-CSF, granulocyte-colony stimulating factor; EBV, Epstein–Barr virus.

**Table 4 cancers-16-01537-t004:** Clinical, pathological and molecular features of the new provisional subtypes of intrahepatic cholangiocarcinoma ^(a)^.

Subtype	Relative Frequency	Clinical Features	Pathological Features	Molecular Features	Prognosis ^(b)^	References
Tubulocystic carcinoma of the bile duct	<1%	No distinct findings to date	Cystically dilated tubules with intracystic papillary growth, grossly sponge-like appearance	No distinct findings to date	Unclear	[147,148,149]
Cholangioblastic (solid-tubulocystic or thyroid follicle-like) cholangiocarcinoma	<1%	Younger women (average age, approximately 40 years)	Wide range of morphology, including solid, trabecular, microcystic, follicular, blastemal-like areas	*NIPBL::NACC1* fusion	Unclear	[150,151,152,153,154,155,156,157]
Enteroblastic cholangiocarcinoma	<1%	Extremely rare, especially extrahepatic bile duct	Polygonal cells with tubular and papillary growth or columnar with clear cytoplasm (fetal gut-like)	Loss of *CDKN2A* and loss of chromosome 18	Worse	[158,159,160,161,162,163,164,165]
Micropapillary carcinoma	<1%	No distinct findings to date	Micropapillary clusters without fibrovascular cores	No distinct findings to date	Worse	[171,172,173]
Acinar cell carcinoma	<1%	No distinct findings to date	Uniform round nuclei with moderate amount of granular, eosinophilic to amphophilic cytoplasm containing zymogen granules	No distinct findings to date	Better	[176,177,178]

^(a)^ Provisional subtypes of ICCA are not included in the 2019 WHO classification (5th edition) of liver tumors ^(b)^ compared with the prognosis of conventional intrahepatic cholangiocarcinoma.

**Table 5 cancers-16-01537-t005:** Immunohistochemical markers for distinguishing intrahepatic cholangiocarcinoma from metastatic carcinomas.

Tumor Origin	Markers	NOTE	References
Adrenocortical	SF1, inhibin, Mart-1/Melan-A, synaptophysin, calretinin	SF1 is the most reliable biomarker with which to confirm the cortical origin	[189,190,191]
Bile duct	CK7, CK19, CA19-9, CEA	CK19 and CA19-9 show the highest sensitivity in ICCA	[192,193,194,195]
Breast	ER, PR, GCDFP-15, mammaglobin, GATA3, TRPS1	TRPS1 is a highly sensitive and specific marker for breast carcinoma; however, it causes significant staining in urinary bladder and prostate cancer	[192,196,197,198]
Colorectal	CK20, MUC2, CDX2, SATB2	An antibody panel of CK7, CK20, CDX2, SATB2, and MUC2 can aid in the distinction between ICCA and metastatic colorectal adenocarcinomas	[192,199,200]
Germ cell	PLAP, OCT4, hCG, CD30, SALL4, LIN28, CD117, D2-40, SOX2, AFP, glypican-3	PLAP, OCT4, hCG, and CD30 are commonly used markers for detecting germ cell tumors	[201,202,203,204,205]
Hepatocellular	Hep Par-1, arginase-1, CD10, polyclonal CEA, AFP	Arginase-1 is a highly sensitive and specific marker for HCC and is better than Hep Par-1 in poorly differentiated HCC	[196,199,206]
Lung	CK7, TTF-1, Napsin A	TTF-1 is widely used as a specific marker for pulmonary adenocarcinoma but can be expressed in neuroendocrine tumors, papillary thyroid carcinoma, and some female genital tract carcinomas	[196,199,207,208]
Mesothelial	Calretinin, D2-40 (podoplanin), WT1, CK5/6	It is advisable to use panels of positive mesothelial markers (three or four) and negative antibodies	[196,209,210,211]
Neuroendocrine	Chromogranin A, synaptophysin, CD56, INSM1	Chromogranin A is more specific than synaptophysin; INSM1 is a recently discovered, useful neuroendocrine marker in primary and metastatic NETs	[192,199,212,213]
Ovarian	PAX8, WT1, Napsin A, CA125, PR, ER	A panel consisting of PAX-8, WT1, and CA125 is useful for the diagnosis of primary ovarian carcinoma	[214,215]
Pancreatic duct	CK7, CK19, SMAD4, p16	CK7 and CK19 are usually positive in ICCA and PDA; loss of SMAD4 expression is more common in PDA than in ICCA	[192,199,216,217,218]
Prostate	PSA, PSAP, PSMA, NKX3.1, P504S (AMACR),	PSA is a specific marker for prostatic carcinoma but approximately 10% of high-grade prostatic carcinoma are negative for PSA; other prostatic-specific markers such as NKX3.1 are useful for confirming the diagnosis	[196,219,220,221,222]
Renal	CD10, PAX2, PAX8, vimentin, CAIX, RCC marker	PAX-8 is expressed in a wide range of tumors and must be used as a part of diagnostic panels, including CAIX and PAX2	[196,205,223]
Squamous	CK5/6, p40, p63	CK5/6, p40, and p63 are useful for confirming squamous cell carcinoma	[224,225]
Urothelial	GATA3, p63, uroplakin, CK5/6	GATA3, p63, and uroplakin are most useful for confirming metastatic urothelial carcinoma	[196,226,227,228]

SF1, steroidogenesis factor 1; CK, cytokeratin; ICCA, intrahepatic cholangiocarcinoma; CA19-9, carbohydrate antigen 19-9; CEA, carcinoembryonic antigen; ER, estrogen receptor; PR, progesterone receptor; CDX2, caudal type homeobox 2; SATB2, special AT-rich sequence–binding protein 2; TTF1, thyroid transcription factor 1; ER, estrogen receptor; PR, progesterone receptor; GCDFP-15, gross cystic disease fluid protein 15; GATA3, GATA binding protein 3; TRPS1, ticho-rino-palangeal syndrome type 1; transcriptional repressor; SATB2, special AT-rich sequence-binding protein 2; PLAP, placental alkaline phosphatase; OCT4, octamer-binding transcription factor 4; hCG, human chorionic gonadotropin; SALL4, spalt-like transcription factor 4; SOX2, SRY-Box transcription factor 2; AFP, alpha-fetoprotein; Hep Par-1, hepatocyte paraffin-1; WT1, Wilms tumor 1; INSM1, insulinoma-associated protein 1; SMAD4, SMAD Family Member 4; PSA, prostate-specific antigen; PSAP, prostate-specific acid phosphatase; PSMA, prostate-specific membrane antigen; NKX3.1, NK3 homeobox 1; AMACR, alpha-methylacyl-CoA racemase; PAX2, paired box gene 2; CAIX, carbonic anhydrase IX; RCC marker, renal cell carcinoma marker.

## Data Availability

All data were included in the manuscript.

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
