# Peer review of "Recent Advances in Pathology of Intrahepatic Cholangiocarcinoma"

_cancers, 2024, doi:10.3390/cancers16081537_

Round 1

Reviewer 1 Report

Comments and Suggestions for Authors

ICCA is a primary carcinoma of the liver with increasing significance and major pathogenic challenges. This review details the subtypes and histopathological features of ICCA. The authors also provide a comprehensive pathological diagnostic approach of ICCA. Overall, the study was informative and of good reference value. There are some comments taken for consideration.

1.      According to the research by the Liver Cancer Study Group of Japan (PMID: 26430782), ICCA was categorized into four types including the three types mentioned in Part 8 and another type called “the mixed pattern”. The authors should supplement this type because it carries a significantly higher risk of death than the MF pattern.

2.      In Part 8.4, the authors introduce two premalignant lesions of ICCA including BilIN and IPNB. However, according to the report (PMID: 28261592), there appears to be another precursor lesion "intraductal tubulo-papillary neoplasm (ITPN)" that has been missed.

3.      Pathology of ICCA represents a relatively limited issue, so the authors should try not to miss some important reviews or expert consensus (PMID: 38161496/28261592).

4.      Some parts are not very well structured, leading to some ambiguity. For instance, is the content of “8.3 Histological Grading” only applicable to the conventional intrahepatic cholangiocarcinoma? In addition, the authors mention premalignant lesions only in Part 8. Are there any reports of premalignant lesions associated with unconventional ICCAs?

5.      Please standardize the abbreviation of intrahepatic cholangiocarcinoma as “ICCA” or “ICC”.

6.      Please add the source of specimens and pathology slides of ICCA shown in the manuscript.

Comments on the Quality of English Language

No comments.

Author Response

ICCA is a primary carcinoma of the liver with increasing significance and major pathogenic challenges. This review details the subtypes and histopathological features of ICCA. The authors also provide a comprehensive pathological diagnostic approach of ICCA. Overall, the study was informative and of good reference value. There are some comments taken for consideration.

  1. According to the research by the Liver Cancer Study Group of Japan (PMID: 26430782), ICCA was categorized into four types including the three types mentioned in Part 8 and another type called “the mixed pattern”. The authors should supplement this type because it carries a significantly higher risk of death than the MF pattern.

-> We appreciate your suggestion. We have described as follows.

“ICCAs are grossly classified into four types: mass-forming (MF), periductal infiltrating (PI), intraductal growth (IG), and mixed type. … Mixed MF and PI types have often a more unfavorable prognosis than those with other types of ICCA [68].”

  1. Sakamoto, Y.; Kokudo, N.; Matsuyama, Y.; Sakamoto, M.; Izumi, N.; Kadoya, M.; Kaneko, S.; Ku, Y.; Kudo, M.; Takayama, T.; et al. Proposal of a new staging system for intrahepatic cholangiocarcinoma: Analysis of surgical patients from a nationwide survey of the Liver Cancer Study Group of Japan. Cancer 2016, 122, 61-70, doi:10.1002/cncr.29686.

  1. In Part 8.4, the authors introduce two premalignant lesions of ICCA including BilIN and IPNB. However, according to the report (PMID: 28261592), there appears to be another precursor lesion "intraductal tubulo-papillary neoplasm (ITPN)" that has been missed.

-> Thank you for your comments. We have described as follows.

“Large duct ICCA can develop from three different types of premalignant intraductal lesions: biliary intraepithelial neoplasia (BilIN), intraductal papillary neoplasm of the bile ducts (IPNB), and intraductal tubulo-papillary neoplasm of the bile ducts (ITPN) [30,31].”

”BilIN, IPNB, and ITPN are precursors of the large duct ICCA,”

“ITPN is characterized as a preinvasive, mass-forming intraductal neoplasm of the intra- or extrahepatic bile ducts, consisting predominantly of nonmucinous tubular structures with and without sheet-like growth (≥70% of the neoplasm) and with no or only minimal papillary growth [79,80]. The majority of ITPNs are found to be associated with invasive carcinoma at the time of diagnosis.”

  1. Pehlivanoglu, B.; Adsay, V. Intraductal tubulopapillary neoplasms of the bile ducts: identity, clinicopathologic characteristics, and differential diagnosis of a distinct entity among intraductal tumors. Hum. Pathol. 2023, 132, 12-19, doi:10.1016/j.humpath.2022.07.019.
  2. Schlitter, A.M.; Jang, K.T.; Klöppel, G.; Saka, B.; Hong, S.M.; Choi, H.; Offerhaus, G.J.; Hruban, R.H.; Zen, Y.; Konukiewitz, B.; et al. Intraductal tubulopapillary neoplasms of the bile ducts: clinicopathologic, immunohistochemical, and molecular analysis of 20 cases. Mod. Pathol. 2015, 28, 1249-1264, doi:10.1038/modpathol.2015.61.

  1. Pathology of ICCA represents a relatively limited issue, so the authors should try not to miss some important reviews or expert consensus (PMID: 38161496/28261592).

-> We appreciate your suggestion. We have described as follows in section 12 of Future Perspectives.

“The current classification of ICCA based on anatomical origin poses several problems, particularly in accurately assessing the epidemiological background, carcinogenesis, and patient outcome [230]. ICCA is highly heterogeneous in terms of cell origin, tissue structure, immunophenotype, and molecular mutations. The criteria for the histological classification of ICCA, the selection of immunohistochemical markers, the detection of molecular targets. and the differential diagnosis of histological subtypes, need to be refined for better application in practice [231].”

  1. Vijgen, S.; Terris, B.; Rubbia-Brandt, L. Pathology of intrahepatic cholangiocarcinoma. Hepatobiliary Surg. Nutr. 2017, 6, 22-34, doi:10.21037/hbsn.2016.11.04.
  2. Wang, H.; Chen, J.; Zhang, X.; Sheng, X.; Chang, X.Y.; Chen, J.; Chen, M.S.; Dong, H.; Duan, G.J.; Hu, H.P.; et al. Expert Consensus on Pathological Diagnosis of Intrahepatic Cholangiocarcinoma (2022 version). J. Clin. Transl. Hepatol. 2023, 11, 1553-1564, doi:10.14218/jcth.2023.00118.

  1. Some parts are not very well structured, leading to some ambiguity. For instance, is the content of “8.3 Histological Grading” only applicable to the conventional intrahepatic cholangiocarcinoma? In addition, the authors mention premalignant lesions only in Part 8. Are there any reports of premalignant lesions associated with unconventional ICCAs?

-> We appreciate your suggestion. No definitive criteria for the histological grading of ICCAs have been established. Histological grading is only applicable to the conventional ICCAs. We have described as follows.

“Premalignant lesions of unconventional ICCAs are not known. Unconventional ICCAs usually develop on the background of a nonbiliary chronic liver disease and cirrhosis”

  1. Please standardize the abbreviation of intrahepatic cholangiocarcinoma as “ICCA” or “ICC”.

-> Thank you for your comments. We standardize the abbreviation of intrahepatic cholangiocarcinoma as “ICCA”.

  1. Please add the source of specimens and pathology slides of ICCA shown in the manuscript.

-> Thank you for your suggestion. The source of the specimen and pathology in the manuscript comes from author’s own files.

Reviewer 2 Report

Comments and Suggestions for Authors

A good review of ICCA, the authors focused on the advances of the pathological findings in the diagnosis of ICCA, I recommend the authors include some prognostic analysis in this review, which tumor types show a poor survival rate.

Author Response

A good review of ICCA, the authors focused on the advances of the pathological findings in the diagnosis of ICCA, I recommend the authors include some prognostic analysis in this review, which tumor types show a poor survival rate.

-> Thank you for your suggestion. There are limited data on survival rates by most rare tumor subtypes. In comparison to the prognosis of conventional intrahepatic cholangiocarcinoma, the survival rate of subtypes was described in text and Table 3 and 4.

Reviewer 3 Report

Comments and Suggestions for Authors

Choi and Thung present a comprehensive overview of iCCA histological subtypes, primarily focusing on the rarer and newer ones. The tables and figures are provided in an organized and helpful manner. As I am not a pathologist nor a clinician, I am limited in my ability to review the bulk of the manuscript that delves into the deep histopathological details of each iCCA subtype. However, overall, I think there may be some missed opportunities to further organize and help the iCCA research field, which I explain in more detail below:

Major comments:

1.      The order of the sections is somewhat confusing, as it begins to talk about small and large duct subtypes before they have even been well defined. I suggest to move section 8 up earlier to be section 2 or 3. Also, it will benefit from additional histopathological detail comparing small to large duct.

2.      It is unclear what exactly is meant by iCCA “conventional” and “subtypes” in Fig. 7. It is particularly confusing, as some people will call small duct and large duct “subtypes” as well. I would be much more comfortable with what is currently being called “subtypes” instead being called “rare subtypes”, while small and large duct are called “conventional subtypes”, if that sounds reasonable to the authors.

a.      Moreover, cholangiocellular and ductal type with malformation are considered “subtypes” of small duct, but are not branching out from under small duct in Fig. 7. This is very confusing.

b.      Are the other rare subtypes completely separate from small and large duct?

c.      Fig. 7, can you clarify what is meant by “metastasis”? From other cancer types? If so please clarify that on the figure.

3.      Could the authors explain more about the relationship between “well, moderate, and poorly differentiated” and the array of conventional and rare subtypes? This is especially interesting in the case of poor differentiation where glandular structures are absent. Is there some way that this can be worked into the tables or figures as well?

4.      Section 3, no one has yet proven that small and large duct arise from distinct cells of origin 100% of the time, which is what the language suggests. It is very likely that some minority of each can have overlapping cells of origin (including, in the mouse, hepatocytes). The language should be more careful about this.

5.      Section 6 and Figure 1, the scheme of molecular division into proliferation and inflammation classes is not widely accepted in the field, as these classifiers have not been readily replicated in other datasets. It is suggested that this paragraph and figure are either removed from the manuscript, or replaced by small duct/large duct signatures defined by e.g. Anderson et al.

Minor comments:

1.      Section 5, radiological features, it is not clear that this is relevant to the manuscript.

2.      Can bile duct adenomas progress to iCCA? If so, what subtype(s)?

3.      TP53 mutations should be mentioned in Section 6 paragraph 2 as being associated with large duct.

4.      I frequently see “anastomosing” to describe iCCA histopathology, but do not see it in this manuscript. Please include some clarifying sentences to incorporate this terminology, and to which subtype(s) it most appropriately applies to.

Author Response

Choi and Thung present a comprehensive overview of iCCA histological subtypes, primarily focusing on the rarer and newer ones. The tables and figures are provided in an organized and helpful manner. As I am not a pathologist nor a clinician, I am limited in my ability to review the bulk of the manuscript that delves into the deep histopathological details of each iCCA subtype. However, overall, I think there may be some missed opportunities to further organize and help the iCCA research field, which I explain in more detail below:

Major comments:

  1. The order of the sections is somewhat confusing, as it begins to talk about small and large duct subtypes before they have even been well defined. I suggest to move section 8 up earlier to be section 2 or 3. Also, it will benefit from additional histopathological detail comparing small to large duct.

-> Thank you for your suggestion. We agree with you. First, in the second paragraph of the section 1 of Introduction, we briefly defined characteristics of small and large duct ICCA.

“According to the World Health Organization (WHO) classification of digestive system tumors (5th edition), ICCAs have two main subtypes; small duct and large duct type [1].”

And risk factors and cell origins of each were described in the following order.

Additional histopathological features comparing small to large duct were described as follows.

“Compared to small duct ICCAs, large duct ICCAs are characterized by the formation of irregular and angulated glands that infiltrate a desmoplastic stroma and abundant cytoplasmic and intraluminal mucin production. In poorly differentiated large duct ICCAs, scattered nests of pleomorphic cancer cells are present.”

  1. It is unclear what exactly is meant by iCCA “conventional” and “subtypes” in Fig. 7. It is particularly confusing, as some people will call small duct and large duct “subtypes” as well. I would be much more comfortable with what is currently being called “subtypes” instead being called “rare subtypes”, while small and large duct are called “conventional subtypes”, if that sounds reasonable to the authors.

-> Thank you for your comment. We changed “subtypes” to “rare subtypes”.

  1. Moreover, cholangiocellular and ductal type with malformation are considered “subtypes” of small duct, but are not branching out from under small duct in Fig. 7. This is very confusing.

-> Thank you for your suggestion. In Figure 7, we changed “subtype of ICCA” to rare subtype of ICCA and described as follows in note of Figure 7.

a)CLLCA and b) ICDPM are considered small duct type of conventional ICCA.”

  1. Are the other rare subtypes completely separate from small and large duct?

-> Thank you for your suggestion. Other rare subtypes are not yet completely separated from the small and large duct ICCA types.

  1. Fig. 7, can you clarify what is meant by “metastasis”? From other cancer types? If so please clarify that on the figure.

-> Thank you for your suggestion. We changed “metastasis” to “metastatic carcinoma”.

  1. Could the authors explain more about the relationship between “well, moderate, and poorly differentiated” and the array of conventional and rare subtypes? This is especially interesting in the case of poor differentiation where glandular structures are absent. Is there some way that this can be worked into the tables or figures as well?

-> Thank you for your suggestion. It is not easy to explain the relationship between “well, moderate, and poorly differentiated” and the array of conventional and rare subtypes. The histologic grading depends on morphology and the extent of gland formation. Generally, grading of a neoplasm demands a morphologic variation within a given tumor. For example, because signet-ring cell carcinoma has little variation, there is no practical way to grade signet-ring cell carcinoma. Poorly differentiated carcinoma without glandular structures is designated for undifferentiated carcinoma.

  1. Section 3, no one has yet proven that small and large duct arise from distinct cells of origin 100% of the time, which is what the language suggests. It is very likely that some minority of each can have overlapping cells of origin (including, in the mouse, hepatocytes). The language should be more careful about this.

-> Thank you for your suggestion. We described as follows.

“The cellular origins of ICCA are very heterogeneous and have not sufficiently proven in all cases.”

  1. Section 6 and Figure 1, the scheme of molecular division into proliferation and inflammation classes is not widely accepted in the field, as these classifiers have not been readily replicated in other datasets. It is suggested that this paragraph and figure are either removed from the manuscript, or replaced by small duct/large duct signatures defined by e.g. Anderson et al.

-> Thank you for your suggestion. We agree with you. The paragraph and Figure 1 were removed.

Minor comments:

  1. Section 5, radiological features, it is not clear that this is relevant to the manuscript.

-> Thank you for your comments. We included radiological features because they are important for the diagnosis of ICCA and the differential diagnosis.

  1. Can bile duct adenomas progress to iCCA? If so, what subtype(s)?

-> Thank you for your comments. We describe as follows.

 “The malignant transformation in bile duct adenoma is considered to be extremely low [81]. The presence of a high frequency of BRAF V600E mutations in bile duct adenomas suggests that they are true neoplasms and that they may be important precursors for the subset of ICCA that harbor BRAF mutations [82].”

  1. Hasebe, T.; Sakamoto, M.; Mukai, K.; Kawano, N.; Konishi, M.; Ryu, M.; Fukamachi, S.; Hirohashi, S. Cholangiocarcinoma arising in bile duct adenoma with focal area of bile duct hamartoma. Virchows Arch. 1995, 426, 209-213, doi:10.1007/bf00192644.
  2. Pujals, A.; Amaddeo, G.; Castain, C.; Bioulac-Sage, P.; Compagnon, P.; Zucman-Rossi, J.; Azoulay, D.; Leroy, K.; Zafrani, E.S.; Calderaro, J. BRAF V600E mutations in bile duct adenomas. Hepatology 2015, 61, 403-405, doi:10.1002/hep.27133.

  1. TP53 mutations should be mentioned in Section 6 paragraph 2 as being associated with large duct.

-> Thank you for your comments. We described as follows.

TP53 mutations are associated with large duct ICCA.”

  1. I frequently see “anastomosing” to describe iCCA histopathology, but do not see it in this manuscript. Please include some clarifying sentences to incorporate this terminology, and to which subtype(s) it most appropriately applies to.

-> Thank you for your suggestion. We described it as follow in section 8.1..

“anastomosing trabecular”

In addition, the term “anastomosing” applied to the cholangiolocarcinoma subtype section as follows. “The tumor is characterized by anastomosing cords…”

Round 2

Reviewer 3 Report

Comments and Suggestions for Authors

The authors have addressed some of my issues, but several remain unsatisfactory.

1.-> Thank you for your suggestion. We agree with you. First, in the second paragraph of the section 1 of Introduction, we briefly defined characteristics of small and large duct ICCA.

“According to the World Health Organization (WHO) classification of digestive system tumors (5th edition), ICCAs have two main subtypes; small duct and large duct type [1].”

And risk factors and cell origins of each were described in the following order.

Additional histopathological features comparing small to large duct were described as follows.

“Compared to small duct ICCAs, large duct ICCAs are characterized by the formation of irregular and angulated glands that infiltrate a desmoplastic stroma and abundant cytoplasmic and intraluminal mucin production. In poorly differentiated large duct ICCAs, scattered nests of pleomorphic cancer cells are present.”

>>>I like the new paragraph, but it should be in Section 1. Readers who don't know anything about cholangio need to get an idea of what exactly is meant by small and large duct before they continue reading.

2. -> Thank you for your suggestion. Other rare subtypes are not yet completely separated from the small and large duct ICCA types.

>>> This needs to be stated in the opening paragraph of Section 9.

3. -> Thank you for your suggestion. It is not easy to explain the relationship between “well, moderate, and poorly differentiated” and the array of conventional and rare subtypes. The histologic grading depends on morphology and the extent of gland formation. Generally, grading of a neoplasm demands a morphologic variation within a given tumor. For example, because signet-ring cell carcinoma has little variation, there is no practical way to grade signet-ring cell carcinoma. Poorly differentiated carcinoma without glandular structures is designated for undifferentiated carcinoma.

>>> This needs to actually be addressed in the manuscript, not just in response to the reviewer. Which rare subtypes have glandular formation and which don't?

4. -> Thank you for your suggestion. We described as follows.

“The cellular origins of ICCA are very heterogeneous and have not sufficiently proven in all cases.”

>>> The new sentence should be removed. After thinking more, it would be sufficient to simply change the sentence "The two main histological subtypes of ICCAs, the small and large duct types, arise from different cell types." to "The two main histological subtypes of ICCAs, the small and large duct types, ARE THOUGHT TO arise from different cell types [21].

5. -> Thank you for your comments. We described as follows.

TP53 mutations are associated with large duct ICCA.”

>>> Remove this new sentence and simply add TP53 into the list with KRAS and SMAD4 in the preceding sentence.

6.

-> Thank you for your suggestion. We described it as follow in section 8.1..

“anastomosing trabecular”

In addition, the term “anastomosing” applied to the cholangiolocarcinoma subtype section as follows. “The tumor is characterized by anastomosing cords…”

>>> I am looking for something more than this. "Anastamosing" is widely used in the literature including when describing cholangiolar. There should be at least a couple sentences introducing and defining the term, and then describing which subtypes it applies to.

Author Response

Authors’ response to Reviewer 3

The authors have addressed some of my issues, but several remain unsatisfactory.

1.

-> Thank you for your suggestion. We agree with you. First, in the second paragraph of the section 1 of Introduction, we briefly defined characteristics of small and large duct ICCA.

“According to the World Health Organization (WHO) classification of digestive system tumors (5th edition), ICCAs have two main subtypes; small duct and large duct type [1].”

And risk factors and cell origins of each were described in the following order.

Additional histopathological features comparing small to large duct were described as follows.

“Compared to small duct ICCAs, large duct ICCAs are characterized by the formation of irregular and angulated glands that infiltrate a desmoplastic stroma and abundant cytoplasmic and intraluminal mucin production. In poorly differentiated large duct ICCAs, scattered nests of pleomorphic cancer cells are present.”

>>>I like the new paragraph, but it should be in Section 1. Readers who don't know anything about cholangio need to get an idea of what exactly is meant by small and large duct before they continue reading.

->Thank you for your comments. We agree with your consideration for readers. We described it in the new paraphrase of Section 1 as follows.

“According to the World Health Organization (WHO) classification of digestive system tumors (5th edition), ICCAs have two main subtypes: small duct and large duct types [1]. Small duct type ICCA occurs in the peripheral parts of the liver, also called peripheral type. The prevalence of this subtype is regionally dependent and accounts for approximately 40%–90% of ICCAs. The large duct type ICCA arises in the larger intrahepatic bile ducts near the hepatic hilum, close to the right and left hepatic ducts, and is also called perihilar type. These two types differ in their etiologies, clinical behavior, and pathological features and have different genetic alterations [9,10]. In the future, it will be necessary to incorporate these findings into clinical research and study processes.”

  1.  

-> Thank you for your suggestion. Other rare subtypes are not yet completely separated from the small and large duct ICCA types.

>>> This needs to be stated in the opening paragraph of Section 9.

->Thank you for your comments. We described it in the opening paragraph of Section 9 as follows.

“Other rare subtypes are not yet completely separated from the small and large duct ICCA types.”

3.

-> Thank you for your suggestion. It is not easy to explain the relationship between “well, moderate, and poorly differentiated” and the array of conventional and rare subtypes. The histologic grading depends on morphology and the extent of gland formation. Generally, grading of a neoplasm demands a morphologic variation within a given tumor. For example, because signet-ring cell carcinoma has little variation, there is no practical way to grade signet-ring cell carcinoma. Poorly differentiated carcinoma without glandular structures is designated for undifferentiated carcinoma.

>>> This needs to actually be addressed in the manuscript, not just in response to the reviewer. Which rare subtypes have glandular formation and which don't?

->Thank you for your comments. We described as follows.

“The histologic grading depends on the morphology and extent of gland formation. Generally, grading a neoplasm requires morphologic variation within a given tumor. For example, because signet-ring cell carcinoma has little variation, there is no practical way to grade this type of carcinoma. Rare ICCA subtypes with no gland formation include lymphoepithelioma-like carcinoma (LELCCA), sarcomatous ICCA, and acinar cell carcinoma, whereas ICCA subtypes with glandular formation include adenosquamous carcinoma.”

  1.  

-> Thank you for your suggestion. We described as follows.

“The cellular origins of ICCA are very heterogeneous and have not sufficiently proven in all cases.”

>>> The new sentence should be removed. After thinking more, it would be sufficient to simply change the sentence "The two main histological subtypes of ICCAs, the small and large duct types, arise from different cell types." to "The two main histological subtypes of ICCAs, the small and large duct types, ARE THOUGHT TO arise from different cell types [21].

->Thank you for your comments. We removed the new sentence. We described as follows.

 “The two main histological subtypes of ICCAs, the small and large duct types, are though to arise from different cell types [21].”

  1.  

-> Thank you for your comments. We described as follows.

“TP53 mutations are associated with large duct ICCA.”

>>> Remove this new sentence and simply add TP53 into the list with KRAS and SMAD4 in the preceding sentence.

->Thank you for your suggestion. We removed this new sentence and simply added TP53 as follows.

mutations in KRAS, SMAD4, and TP53…”.

6.

-> Thank you for your suggestion. We described it as follow in section 8.1.

“anastomosing trabecular”

In addition, the term “anastomosing” applied to the cholangiolocarcinoma subtype section as follows. “The tumor is characterized by anastomosing cords…”

>>> I am looking for something more than this. "Anastomosing" is widely used in the literature including when describing cholangiolar. There should be at least a couple sentences introducing and defining the term, and then describing which subtypes it applies to.

->Thank you for your comments. We described as follows.

“An anastomosing pattern refers to cancer cells forming a network-like structure that branches and reconnects similarly to cholangiolar structures. This growth pattern is commonly present in the ICCA subtype, particularly in cholangiolocarcinoma.”
